# Protein Disulfide Isomerase FgEps1 Is a Secreted Virulence Factor in *Fusarium graminearum*

**DOI:** 10.3390/jof9101009

**Published:** 2023-10-12

**Authors:** Kouhan Liu, Xintong Wang, Ying Li, Yifeng Shi, Yanyan Ren, Aolin Wang, Bingjie Zhao, Peng Cheng, Baotong Wang

**Affiliations:** State Key Laboratory of Crop Stress Biology for Arid Areas, College of Plant Protection, Northwest A&F University, Xianyang 712100, China; liukouhan6@nwafu.edu.cn (K.L.); xintongwang@nwafu.edu.cn (X.W.); liying2018@nwafu.edu.cn (Y.L.); yfshi@nwafu.edu.cn (Y.S.); nwafuryy@nwafu.edu.cn (Y.R.); orrinw@126.com (A.W.); 15848154728@163.com (B.Z.)

**Keywords:** *Fusarium graminearum*, protein disulfide isomerase *FgEps1*, fungal growth, virulence factor

## Abstract

Protein disulfide isomerase (PDI) is a member of the thioredoxin (Trx) superfamily with important functions in cellular stability, ion uptake, and cellular differentiation. While PDI has been extensively studied in humans and animals, its role in fungi remains relatively unknown. In this study, the biological functions of FgEps1, a disulfide bond isomerase in the fungal pathogen *Fusarium graminearum*, were investigated. It was found that *FgEps1* mutation affected nutritional growth, asexual and sexual reproduction, and stress tolerance. Additionally, its deletion resulted in reduced pathogenicity and impaired DON toxin biosynthesis. The involvement of *FgEps1* in host infection was also confirmed, as its expression was detected during the infection period. Further investigation using a yeast signal peptide secretion system and transient expression in *Nicotiana benthamiana* showed that *FgEps1* suppressed the immune response of plants and promoted infection. These findings suggest that virulence factor *FgEps1* plays a crucial role in growth, development, virulence, secondary metabolism, and host infection in *F. graminearum.*

## 1. Introduction

Fusarium head blight (FHB) is a major disease of small grain cereals worldwide and caused by *Fusarium graminearum* [1]. The pathogen typically invades the host through secondary spore infections, and its sexual reproduction produces perithecia that act as a primary source of infection in the following year [2]. This highly destructive fungus leads to unsaturated wheat seeds, significantly reduced germination and emergence rates, and varying degrees of yield reduction, sometimes exceeding 50% or resulting in direct crop failure [3,4]. Additionally, *F. graminearum* produces several mycotoxins, such as deoxynivalenol (DON) and zearalenone (ZEA), which pose serious risks to the safety of humans and animals [5]. The biosynthesis of DON occurs through the activity of the *Tri* gene cluster [6]. The management of FHB remains challenging, primarily due to the lack of germplasm resources that are fully resistant or immune to *F. graminearum*. The current control method involves using azole fungicides during the flowering period in wheat, which is expensive and time-consuming. Therefore, a better understanding of pathogen–host interaction may allow for novel engineered resistance to be designed in the host. To penetrate and colonize different plant tissues and cause disease, fungi must disable the plant’s defense mechanisms. The secretion of fungal proteins, known as effectors, is crucial for pathogenesis, as these proteins modulate plant immunity and facilitate infection [7]. Intracellular immune receptors can recognize specific fungi effectors or effector activities, inducing effector-triggered immunity (ETI). A range of effectors have been identified to date, most of which are small, secreted proteins [8]. The target genes of fungal effectors serve a variety of functions, including acting as transcription factors, protein kinases and compounds involved in plant defense, and metabolic pathways, as well as signaling [9,10,11,12]. Avr2, an effector protein from the tomato pathogen *Cladosporium fulvum* (syn. Passalora fulva), can decrease the pathogenicity and virulence of the pathogen to tomatoes when silenced [13,14].

In *Ustilago maydis*, the effector Tin2 is induced during the pre-infection period, hindering plant metabolism and thereby promoting pathogen invasion. A deficiency in Tin2 impairs the ability of *U. maydis* to extract nutrients from the host through vascular tissues [15]. During the secretion of effector proteins, protein modifications such as disulfide bonding, glycosylation, or acetylation play a crucial role in preventing protein degradation or structural variations. These modifications provide a stable structure for the polypeptide chains, enhancing thermodynamic stability and enabling better adaptation to the external environment.

Fungal effector proteins are rich in cysteine amino acids, and the covalent bonds between different cysteine amino acid residues (known as disulfide bonds) play a significant role in the stability of the correctly folded state of proteins [16]. The formation of disulfide bonds is catalyzed by a series of redox enzymes, and protein disulfide isomerase (PDI) is increasingly recognized for its catalytic role [17]. PDI is a member of the thioredoxin (Trx) superfamily and has various intracellular functions, such as contributing to cellular stabilization, ion uptake, and cellular differentiation [18]. Further research has revealed that PDI not only catalyzes the redox of protein disulfide bonds and heterodimerization [19] but also has molecular chaperone, Ca^2+^, and Cu^2+^ binding activities and participates in metabolism [20,21,22]. In *Saccharomyces cerevisiae*, the whole genome sequence has revealed a total of five PDI genes, namely *PDI1*, *EUG1*, *MPD1*, *MPD2*, and *EPS1* [23]. Studies have shown that overexpression of the other four genes, excluding *PDI1*, can partially rescue the growth deficiency resulting from PDI deletion [24,25]. PDI also plays a crucial role in host–pathogen interactions [26]. For instance, in *Phytophthora parasitica*, a conserved effector protein PpPDI1 induces cell necrosis in *Nicotiana benthamiana*. Mutants of PpPDI1 affect haustoria structure during infection, and PpPDI1-EGFP transformants enhance the formation of tumor-like structures and exhibit increased virulence against *N. benthamiana* [27]. In parasitic *Leishmania major*, the *LmPDI* gene shows high levels of expression during the infection stage of mice, raising the possibility that *LmPDI* expression affects the degree of infection and may be a pathogenic factor [28]. HsPDI is an effector protein that interferes with the interaction between the nematode species *Heterodera schachtii* and the host plant (sugar beet) by inhibiting the accumulation of reactive oxygen species in the host [29]. In a recent study, *MgPDI2*, a typical PDI from the nematode species *Meloidogyne graminicola*, was found to be involved in the pathogenicity of *M. graminicola*. Furthermore, *MgPDI2* induces cell death in *N. benthamiana*, indicating that it may play a vital role in the pathogenicity of *M. graminicola* [30]. In conclusion, PDI, as a conserved protein, plays an essential role in pathogen invasion and host–pathogen interactions. PDI has been extensively studied in humans and animals, and inhibition of PDI gene family members can inhibit the replication of several viruses, such as HIV. Nonspecific inhibition of PDI activity suppressed the PDI-mediated redox environment of plasma membranes and therefore interfered with HIV envelope protein-directed cell fusions. [31]. However, the role of PDI family members in the pathogenicity of *F. graminearum* remains largely unknown.

In this study, we detail the biological functions of *FgEps1* in *F. graminearum* and present direct evidence of its involvement in its pathogenicity. The deletion of *FgEps1* caused dense mycelium, impairment of asexual and sexual reproduction, and an enhanced susceptibility to various stress factors. Most notably, the deletion of *FgEps1* led to decreased pathogenicity and reduced levels of the DON toxin in *F. graminearum*. Moreover, we explored the signal peptide secretion function of *FgEps1* and assessed the expression of *FgEps1* during the different stages of infection. Overexpression of *FgEps1* in *N. benthamiana* resulted in an accumulation of callose and reactive oxygen species (ROS). These findings emphasize the critical role of *FgEps1* in *Fusarium graminearum* infestation of wheat.

## 2. Materials and Methods

### 2.1. Fungal Strains, Culture Conditions, and Plant Materials

The wild-type *F. graminearum* strain PH-1 (NRRL 31084) was provided by the Wheat Pathogenic Fungal Surveillance and Disease Resistance Genetics Laboratory at Northwest A&F University for conservation. The fungus was cultured on solid potato dextrose agar medium (PDA), complete medium (CM), and trace element minimum medium (MM) at 25 °C. The wheat susceptible cultivar “Xiaoyan 22” was used as the susceptible cultivar. *N. benthamiana* was grown in a constant-temperature in vivo incubator with alternating dark/light cycles of 16/8 h at 23 ± 1 °C and 70% relative humidity.

### 2.2. Bioinformatics Analysis

The full-length base sequence of *FgEps1* (FGSG_06174) was amplified from the cDNA of the wild-type PH-1 using specifically designed primers (listed in Appendix A), and multiple sequence alignment was performed using DNAMAN (Lynnon Biosoft, San Ramon, CA, USA). Phylogenetic analysis of Eps1 proteins in different fungi was conducted using MEGA7 software (version 5.0, Mega Limited, Auckland, New Zealand). The phylogenetic tree was constructed using the maximum likelihood method and 1000 bootstrap replicates (Appendix A). The potential signal peptide was predicted using SignalP 5.0 (http://www.cbs.dtu.dk/services/SignalP/index.php (accessed on 18 August 2022).

### 2.3. Knockout of Target Gene and Construction of Complemented Strains

The mutants were created using the double-joint (DJ) PCR approach [32] and polyethyleneglycol (PEG)-mediated protoplast transformation [33]. Three transformants were isolated by a single spore, and DNA was extracted for preliminary tests. The primers used to amplify the flanking sequences of the gene are listed in Appendix A. The open reading frame (ORF) of FgEps1 was replaced with a hygromycin (hyg) resistance cassette (Appendix A). To confirm that the phenotype of FgEps1 was caused by the disruption of the gene, the DNA fragment carrying the native promoter and the ORF of FgEps1 was amplified using the primer pair listed in Appendix A. The relative expression levels of FgEps1 and four pairs of specific primers (ID-F/HYC-R, ID-R/HYC-F, ID-F/ID-R, and HYC-F/HYC-R) at the transcriptional level were used to confirm that FgEps1 was successfully knocked out (Appendix A). When constructing complementary transformants, the complementary fragments were first amplified using characteristic primers (Appendix A). The resulting PCR products were co-transformed with Xho-digested pYF11-GFP-Gen vector into yeast cells XK1-25 using a yeast transformation kit (MP Biomedicals, Solon, OH, USA) to generate the recombined pYF11-FgEps1-GFP-Gen plasmid [34,35]. This plasmid was then extracted from the yeast XK1-25 transformant using a yeast plasmid extract kit (Vazyme, Nanjing, China) and transferred into Escherichia coli strain DH5α to propagate the plasmids. Finally, the recombinant plasmid was transformed into the deletion mutant Δ*FgEps1* to generate the complemented strains via random insertion. For selective growth of complemented transformants, PDA medium supplemented with G418 sulfate (100 mg/L) was used [36]. Deletion candidates and the complemented strain Δ*FgEps1*-C were identified through PCR assays with relevant primers.

### 2.4. Nutritional Growth Test

The activated wild-type strain PH-1, deletion mutant, and complemented strains were grown on PDA, CM, and MM solid medium, respectively, for 3 days. The average growth rate was calculated, and three biological replicates were performed. The sparseness and bifurcation of the marginal mycelial ends of each strain were observed under a 20× electron microscope. Additionally, spore production, the number of spore diaphragms, and spore germination rates were assessed for each strain, and three independent biological and technical replicates were conducted. To determine the sensitivity of each strain to stress factors (such as 1 M KCl, 1 M NaCl, and 15 mM H_2_O_2_), cell wall damaging agents (including 0.2 g/L Congo red, 0.75 g/L caffeine, and 0.02% SDS), metal cations (0.2 M MgCl_2_·6H_2_O, 0.5 M CaCl_2_, CuSO_4_·5H_2_O), and fungicides (0.25 ppm phenamacril and 0.25 ppm tebuconazole), activated strains were inoculated in the center of PDA plates containing varying agents and incubated at 25 °C for 3 days to calculate the inhibition rate. The mycelial growth inhibition rate (MGIR) was calculated according to the formula × 100, MGIR = [(N − C)/C] × 100, where C is the diameter of the control colonies, and N is the diameter of the treated colonies. Each experiment was repeated three times independently.

### 2.5. Sexual Reproduction

To test the sexual reproduction of the mutated and complemented strains, they were inoculated in the center of a carrot agar culture dish. Once the aerial mycelium had grown over the culture dish, the aerial mycelium was scraped off, and 500 μL of 2.5% Tween-20 solution was added. The dish was then incubated for 15 days at 25 °C, alternating between dark and light cycles every 12 h [37]. The ascospore shell and ascospore morphology were observed under the light microscope. The experiment was repeated three times.

### 2.6. Mycelium Penetration Test

The activated strains were inoculated on PDA medium lined with cellophane for 3 days at 25 °C. The differences in mycelial penetration of the different strains were observed, and the mycelia on the cellophane were imaged by a light microscope. The experiment was repeated three times.

### 2.7. Pathogenicity Observation and DON Toxin Expression Level Analysis

To analyze the pathogenicity on maize silk, we selected maize at the flowering stage. After removing the maize bracts, we used tweezers to cut the young maize silks to about 10 cm long. Every three roots were placed in a petri dish with sterile filter paper. A circular agar plug with hyphae with a diameter of 9 mm was inoculated in the middle of the young maize silk and incubated at 25 °C with 60% relative humidity for 4 days. The symptoms of the infected maize silk were observed and recorded after 4 days. For the pathogenicity analysis of field wheat spikes, the wild-type strain PH-1, deletion mutant, and complemented strains were cultured on carboxymethylcellulose sodium medium (CMC) for 3 days. The resulting cultures were filtered and centrifuged, and the concentration of conidia was adjusted to 1 × 10^5^ conidium /mL using sterile water. The conidial suspension of each strain was point inoculated in the middle of the wheat spikes using a pipette, and 25 wheat spikes were inoculated with each strain at 20 μL each time. After 15 days, the statistics of disease index was produced (disease index = prevalence × average severity × 100%; average severity = Σ (severity × number of diseased spikes)/total number of diseased spikes investigated). To quantify DON production, infected spikelets of each strain were collected from inoculated wheat heads in the field, and the yield of the DON toxin was determined using the catalytic toxin ELISA kit (Jiangsu Enzyme Immunity Industry Co., Ltd., Yancheng, China).

### 2.8. Tri Gene Cluster Expression Level

In order to analyze the expression level of the *Tri* gene cluster encoding DON synthesis-related enzymes in strain ΔFgEps1, strain PH-1 was used as a control. According to the method of inoculation of wheat spikes in the field, we inoculated 10 wheat spikes and threshed the diseased spikes after 15 days and then ground them into a fine powder using liquid nitrogen. The total RNA was extracted from each RNA sample using RNAiso reagent (TaKaRa Co., Dalian, China). One microgram of each RNA sample was reverse transcribed using a HiScriptIIQRT Super Mix qPCR kit (Vazyme Biotech, Nanjing, China). The expression level of each gene was determined by quantitative real-time PCR using the primers listed in Appendix A (primers that start with “RT”). The actin gene was used as an internal reference to normalize the expression levels of the target genes. Each experiment was repeated three times independently to ensure reproducibility.

### 2.9. Yeast Signal Peptide Secretion

To insert the predicted sequence of FgEps1 signal peptide into the linearized PSUC2t7M13ori (pSUC2) vector for recombination, the pSUC2 vector that contains the sucrose invertase SUC2 gene lacking the start codon and signal peptide [38] was used. The signal peptide sequence of Avr1b was also inserted into the linearized pSUC2 vector for recombination and served as a positive control, while the pSUC2 empty vector was used as a negative control. The resulting pSUC2-Eps1sp vector was then transformed into the yeast strain YYK12. The transformed colonies were screened on CMD-W medium (0.67% YNB, yeast nitrogen base without amino acids; 0.075% tryptophan dropout supplement; 0.1% glucose; 2% sucrose; 2% bacto agar). Positive transformants were selected and cultured on YPRAA medium (2% peptone; 2% raffinose; 2% bacto agar; 2 mg/mL antimycin A; 1% yeast extract) lacking tryptophan. A small number of transformed colonies were then screened on 2,3,5-triphenyltetrazolium chloride (TTC) medium to observe if a red color could be induced in TTC, which would indicate that the FgEps1 signal peptide could direct the secretion of Suc2 to the yeast cell wall.

### 2.10. Induced Tobacco Cell Necrosis Experiment and Callose, ROS Determination

To express the full-length coding sequence of FgEps1 (containing the signal peptide sequence), specific primers (pBin-F/R) were used to insert it into the pBin vector. The resulting construct was then transiently expressed in *N. benthamiana* using Agrobacterium-mediated (GV3101) transient expression. For co-expression experiments, pBin-Bax and pBin-Eps1 were injected into the leaves of *N. benthamiana* at a 1:1 ratio for co-expression, with a concentration of OD = 600 (optical density). Negative and positive controls were also used, including pBin-GFP and pBin-Bax, respectively. After 2–3 days, the leaves of *N. benthamiana* were observed and decolorized using 100% ethanol. Ion leakage from leaf disks was measured to assay cell death. Six leaf disks (1 cm diameter) from agroinfiltrated areas were taken and floated in 5 mL distilled water for 5 h, and the conductivity of the bathing solution was measured using a conductivity meter (FE32 FiveEasy; Mettler-Toledo, Shanghai, China) to yield “value A.” Then the leaf disks were boiled in the bathing solution in sealed tubes for 20 min. When the solution cooled to RT, the conductivity was measured to gain “value B.” Ion leakage was calculated as percent leakage, that is, (value A/value B) × 100. Assays were repeated three times. One gram of each treated leaf was collected, and the levels of callose and reactive oxygen species (ROS) were measured using a kit (Comin Biotech., Suzhou, China). Each experiment was repeated three times for statistical reliability.

### 2.11. Expression Level of FgEps1 during the Period of Infection

A 20 μL (1 × 10^5^ conidium/mL) conidial suspension of the wild-type strain PH-1 was inoculated into a single floret of wheat spikes using a pipet at anthesis. The inoculated spikelets were sampled every 24 h for 5 consecutive days, and the samples were ground with liquid nitrogen. RNA was extracted from the samples using a kit (RNAiso reagent (TaKaRa Co., Dalian, China)) and reverse transcribed to cDNA using the HiScriptII QRT Super Mix qPCR kit (Vazyme Biotech, Nanjing, China). Real-time PCR (RT-qPCR) was then performed to quantify the level of *FgEps1* expression in *F. graminearum*-infected wheat. Actin was used as an internal reference, and all primers are listed in Appendix A (Ex-Eps1-F/R). Roche’s LightCycle 480 II (Roche, Basel, Switzerland) was used for the real-time PCR. Three biological replicates were performed for statistical analysis of the results.

### 2.12. Statistical Analyses

To analyze the statistical differences in pairwise comparisons, Student’s *t*-test was used. For multiple comparisons, analysis of variance (ANOVA) was performed using the one-way ANOVA method, and the least significant difference (LSD) test was used to determine significance of difference. All statistical analyses were performed using the R program version 4.1.1.

## 3. Results

### 3.1. The Deletion of FgEps1 Has an Effect on the Density and Penetration of Mycelium

To investigate the role of *FgEps1* in the nutritional growth and penetration ability of *F. graminearum*, we inoculated wild-type strain PH-1, deletion mutant strain Δ*FgEps1*, and complemented strain Δ*FgEps1*-C on PDA, CM, and MM agar medium at 25 °C for 3 days, respectively. It was found that deletion of *FgEps1* did not affect the nutritional growth of the strain compared to the wild-type strain PH-1 and complemented strain Δ*FgEps1*-C. However, the deletion of *FgEps1* did cause the marginal mycelium to become denser and more curled, and the divergence angle at the end of the mycelia became smaller (Figure 1A). The growth rates of the different strains are shown in Figure 1B. To simulate the penetration of the Δ*FgEps1* strain into the host plants, we inoculated the activated wild-type strain PH-1, deletion mutant Δ*FgEps1*, and complemented strain Δ*FgEps1*-C in the center of PDA medium lined with sterile cellophane and incubated them at 25 °C for 3 days. It was found that the mycelium of the Δ*FgEps1* strain was much denser and produced less pigment than the other strains. The absence of *FgEps1* was observed to have a reduced penetration ability of *F. graminearum* mycelium to some extent on the uncovered cellophane (Figure 1C). In summary, these results suggest that *FgEps1* is involved in the mycelial morphology or phenotype of *F. graminearum*.

### 3.2. FgEps1 Deficiency Can Severely Impede the Asexual and Sexual Reproduction of F. graminearum

To investigate the role of *FgEps1* in asexual and sexual reproduction of *F. graminearum*, the strains were induced to produce asexual conidia in CMC liquid medium and stained with CFW to measure the conidia produced after 3 days. The mutant strain Δ*FgEps1* produced significantly fewer conidia than the wild-type strain PH-1, with only 8.33% of the conidial production of PH-1, and this phenotype was restored in the complemented strain Δ*FgEps1*-C (Figure 2A,B). The absence of *FgEps1* did not significantly affect the number of conidial septa (Appendix A). To test the role of *FgEps1* in spore germination, spore suspension was enriched, and then spore germination was induced in YEPD liquid medium. The spore germination rate of the deletion mutant Δ*FgEps1* was significantly inhibited compared to WT strain PH-1 and the complemented strain Δ*FgEps1-C* after 2, 4, and 8 h after incubation, while there was no difference in spore germination rates between WT strain PH-1 and the complemented strain Δ*FgEps1*-C (Figure 2C).

To test the role of Δ*FgEps1* in sexual reproduction, the strains were inoculated on carrot agar medium and induced with alternating dark/light every 12 h for 30 days. The mediums inoculated with wild-type strain PH-1 and the complemented strain Δ*FgEps1*-C were covered with purple-black perithecia, while the medium inoculated with strain Δ*FgEps1* produced no perithecia (Figure 2D,E). Taken together, these results indicate that the deletion of *FgEps1* significantly affects asexual and sexual reproduction, as well as spore germination rates.

### 3.3. Sensitivity of ΔFgEps1 to Different Stress Factors

Copper ions are important micronutrients for normal plant growth and response to adversity, while H_2_O_2_ accumulation is an important indicator of plant immunity. To investigate the role of *FgEps1* in environmental stress responses, we tested the sensitivity of the Δ*FgEps1* strain to various stress factors. It was found that the Δ*FgEps1* strain showed increased sensitivity to 0.25 ppm phenamacril, 0.2 g/L Congo red, 0.02% SDS, 1 M KCl, 1 M NaCl, 0.2 M MgCl_2_·6H_2_O, 0.5 M CaCl_2_, and 0.75 g/L caffeine, while the Δ*FgEps1* strain showed increased tolerance to 15 mM H_2_O_2_ and CuSO_4_·5H_2_O (Figure 3A,B). Based on our findings, we speculate that *FgEps1* may play a role in the defense of *F. graminearum* against plant immunity.

### 3.4. FgEps1 Has an Essential Contribution in the Pathogenesis of F. graminearum

It was observed that the maize filament spots inoculated with PH-1 and Δ*FgEps1*-C had spread throughout the maize silk, while those inoculated with Δ*FgEps1* showed restrained spots without expansion (Figure 4A). The length of the spots is shown in Figure 4C. Based on the indoor test, a field assay for pathogenicity was conducted. We inoculated the conidia suspension in the middle of wheat heads at the flowering stage and found that scab symptoms appeared first on wheat spikelets inoculated with PH-1 and Δ*FgEps1*-C, which spread rapidly to the entire spikelets 15 days after inoculation. In contrast, mycelium in the glumes inoculated with Δ*FgEps1* did not spread to the whole spikelet stem (Figure 4B). We subsequently sampled the inoculated wheat to calculate the disease index and average severity (Figure 4D). It was found that the disease index and average severity of wheat inoculated with the mutant were significantly lower than those of PH-1 and Δ*FgEps1*-C. These results indicate that the deletion of *FgEps1* contributes to the reduction of pathogenicity. In summary, our results suggest that *FgEps1* plays an important role in the pathogenicity of *F. graminearum* and may be a potential target for controlling wheat scab disease.

### 3.5. FgEps1 Was Involved in DON Biosynthesis and Full Virulence in F. graminearum

The results showed that all 12 *Tri* genes (*Tri1*, *Tri2*, *Tri4*, *Tri5*, *Tri6*, *Tri7*, *Tri8*, *Tri10*, *Tri11*, *Tri12*, *Tri13*, *Tri14*) in the Δ*FgEps1* strain showed different degrees of low expression level compared to PH-1 (Figure 5A). Next, we measured the biosynthetic production of DON toxin in diseased wheat spikes inoculated with strain PH-1, *FgEps1*-C, and Δ*FgEps1*. As expected, the wheat spikes inoculated with Δ*FgEps1* produced less DON toxin than those inoculated with PH-1 and *FgEps1*-C (Figure 5B). Furthermore, we measured the expression level of *FgEps1* during the infection period of *F. graminearum* in wild type PH-1 and found that the expression level of *FgEps1* increased steeply at day 3 and then decreased (Figure 5C). These results suggest that *FgEps1* affects the invasion of *F. graminearum* and the synthesis of DON toxin in wheat. In conclusion, our findings suggest that *FgEps1* may be a potential target for controlling FHB by regulating the expression of the *Tri* gene cluster and inhibiting DON toxin synthesis.

### 3.6. The Signal Peptide of FgEps1 Has a Secretory Function, and Overexpression Can Suppress the Immune Response in Plants

Signal peptides are specific elements of secreted proteins that play an important role in the activity of fungal invasion of hosts. The Signal Peptide Secretion Prediction Tool (https://services.healthtech.dtu.dk/service.php?SignalP-5.0 (accessed on 18 August 2022) was used to predict the signal peptide of FgEps1 (Figure 6A). Specific primers were designed to amplify the full length of the signal peptide (amino acids 1–18), and pSUC2-FgEps1sp was constructed to confirm whether the N-terminal signal peptide of FgEps1 has a secretory function. The sucrose-converting enzyme secretion system of strain YTK12 was used for this purpose [39]. As indicated in Figure 6C, strain YTK12 could not grow on CMD-W medium lacking tryptophan and could grow on YPRAA only with cottonseed sugar as a carbon source. However, the transformed strains, where the recombinant plasmid replaced the yeast sucrose invertase signal peptide with the target signal peptide Fgeps1sp, were able to utilize the raffinose on YPRAA and grow on it. Additionally, these strains induced a TTC red reaction, demonstrating that the N-terminal signal peptide of FgEps1 had a secretory function. To determine the functions of FgEps1, the full-length coding sequence was inserted into the pBin-GFP vector. Using the *Agrobacterium tumefaciens* infiltration system, FgEps1 was co-injected with the cell necrosis-inducing gene Bax into *N. benthamiana* leaves. It was found that Bax-triggered *N. benthamiana* cell death was inhibited by transient expression of FgEps1 (Figure 6B), which was confirmed by quantification of ion leakage (Appendix A). At 48 h, H_2_O_2_ and callose accumulations in *N. benthamiana* leaves were measured. FgEps1 produced significantly less H_2_O_2_ and callose than Bax in the control groups. The co-injection also inhibited the accumulation of H_2_O_2_ and callose caused by Bax (Figure 6D,E).

## 4. Discussion

Protein modifications, such as disulfide bonds, glycosylation, or acetylation, play a crucial role in the secretion of proteins and peptides by pathogens to avoid protein degradation or structural changes. Evidence supports that the formation of disulfide bonds is catalyzed by PDI oxidoreductase [19]. With the continuous development of technology and research, the versatility of PDI family genes has been gradually explored and expanded. Related studies indicate that PDI and protein-like protein disulfide isomerase (PDIL) act as molecular chaperones and also exhibit certain anti-molecular chaperone activities [21,40]. We infer that fungal homologs also retain this function. We speculate that *FgEps1* acts as a cell chaperone during infection and participates in protein transport and localization. [31]. However, so far, studies on the functions of PDI in vivo have mainly focused on plants and medicine, and few studies on plant pathogenic fungi have been reported. Therefore, this study explored the gene functions of *FgEps1*, a member of the PDI family in *F. graminearum*, and identified direct evidence for its involvement in the pathogenicity mechanism of *F. graminearum*.

In *S. cerevisiae*, both wild-type and ΔScEps1 secrete amyloid-prone cystatin and unstable mutant C94A lysozyme. The secretion level of amyloid-prone cystatin in ΔScEps1 is much higher than that in wild-type strains, while the secretion of disulfide-destroyed unstable mutant C94A lysozyme is greatly increased in ΔScEps1. However, the secretion of unstable mutant C94A lysozyme and amyloid-prone cystatin in ΔScEps1 does not maintain their specific activities [41]. Our homology BLAST search indicated that *FgEps1* is a single-copy gene that encodes a disulfide isomerase Eps1 in *F. graminearum*, containing a signal peptide, a low complexity region, and three thioredoxin-related protein domains. Previous studies have shown that the oxidase activity of PDI in yeast is crucial for growth and development [42,43]. We obtained three *FgEps1* deletion mutants with consistent phenotypes and selected one of them for biological phenotype experiments (Appendix A). Interestingly, our study found that the deletion of *FgEps1* did not affect the vegetative growth of *F. graminearum* (Figure 1B) but affected the growth morphology of marginal hyphae (Figure 1A). Therefore, we speculate that *FgEps1* plays a dual function, participating not only in the infection process but also in the fungus’s growth and development. The invasion of pathogens into the host is determined by the ability of the mycelium or infection structures to penetrate the host plant. Through the cellophane penetration simulation infection process, we found that the deletion of *FgEps1* could reduce the penetration ability of mycelium to some extent (Figure 1C). Studies on related species have shown that *F. oxysporum* loses the ability to penetrate cellophane after mutating the *OCHI* gene and that *STE12* gene mutation leads to a reduction in pathogenicity and the inability to form an infection structure on wheat heads in *F. graminearum* [44,45]. Thus, we speculated that the decreased penetration ability of Δ*FgEps1* might affect *F. graminearum*’s colonization. Surprisingly, the deletion of *FgEps1* hindered *F. graminearum*’s colonization in the wheat spikes pathogenicity test (Figure 4B), which was consistent with our hypothesis. However, it only indicated that *FgEps1* was one factor contributing to the virulence defect of *F. graminearum*. Further investigation is necessary to fully elucidate *FgEps1* and other factors’ roles in *F. graminearum*’s virulence.

One possible factor contributing to the reduced virulence of Δ*FgEps1* may be defects in DON toxin biosynthesis. We confirmed this by observing that *FgEps1* was required for full virulence and DON biosynthesis during *F. graminearum* infection and in *FgEps1*-infected wheat spikes (Figure 5B). Additionally, the low expression of 12 *Tri* genes encoding DON biosynthesis (Figure 5A) could also explain the decrease in DON toxin production. *F. graminearum* typically invades the host through secondary infection, so we tested the sexual and asexual reproduction of Δ*FgEps1* and found that Δ*FgEps1* significantly reduced the number of conidia (Figure 3B), while the germination rate of spores was also affected (Figure 2C). Under natural conditions, the primary infection source of FHB typically comes from ascospores produced on various residues after the overwintering of *F. graminearum.* We observed that the ΔFgEps1 strain did not produce sexual perithecia and ascospores (Figure 2D,E), suggesting that *FgEps1* played a crucial role in the sexual reproduction of *F. graminearum* and that deletion of *FgEps1* may result in the loss of sexual reproduction ability. The possibility that *F. graminearum* lacking *FgEps1* would greatly reduce its ability to re-infect the host in subsequent years remains to be verified. To further explore the role of *FgEps1*, we observed the growth and development ofΔ*FgEps1* for three days and calculated the inhibition rate (Figure 3A,B) under different stress factors. Our results showed that the knockout of *FgEps1* increased the sensitivity of *F. graminearum* to most stress factors but increased tolerance to H_2_O_2_ and CuSO_4_·5H_2_O. Copper ions are crucial trace elements for plant growth and resistance to adversity, and the accumulation of H_2_O_2_ is an essential indicator of a plant’s immune status. Based on our experimental findings, we hypothesize that *FgEps1* plays a role in *F. graminearum*’s ability to resist plant immunity and exercise Cu^2+^ binding activity with PDI. During the coevolution of plants and pathogens, a complex interaction relationship develops. Plants have evolved various recognition and resistance mechanisms to organize and limit pathogen infection, while pathogens have developed diverse pathogenic mechanisms to avoid or overcome plant resistance mechanisms [37]. So far, we have only confirmed that *FgEps1* is involved in the pathogenesis of *F. graminearum* but have not yet analyzed its mode of action in this process. Plant pathogens secrete virulence factors into host cells to promote self-infection and colonization [46]. We verified the signal peptide secretion function of FgEps1 using yeast (Figure 6C). Transient expression of secretory protein in *N. benthamiana* is a common method to identify its functions. Non-toxic proteins that are recognized by plants can induce a hypersensitive response (HR) leading to necrosis of plant leaves, while some proteins can inhibit the HR response in plants. Interestingly, our experiment showed that transient expression of FgEps1 inhibited Bax-induced cell death in *N. benthamiana* cells (Figure 6B), indicating that FgEps1 may be involved in suppressing the host’s basic defense response. *F. graminearum* is a hemi-biotrophic vegetative fungus that does not entirely rely on suppressing plant HR response for infection; still, pathogen infection can cause plant immunity, suppress the HR response, and help *F. graminearum* to better infect. Callose accumulation is a typical feature of the PTI (pattern-triggered immunity) response [47], and the ETI (effector-triggered immunity) response causes the burst of ROS (reactive oxygen species) [48]. To further clarify whether FgEps1 regulates the basic immune response of host cells, we measured the accumulation of H_2_O_2_ and callose in the model plant *N. benthamiana* and found that H_2_O_2_ and callose produced by FgEps1 were significantly reduced compared to Bax (Figure 6D,E). Although the plant defense system is conserved in monocots and dicots, it is meaningful to determine whether *FgEps1* can inhibit the immune response of *F. graminearum* in host plants such as wheat.

In summary, our study found that *FgEps1* plays a crucial role in the various aspects of the life cycle of *F. graminearum*, including asexual growth, sexual reproduction, asexual reproduction, pathogenicity, and DON toxin synthesis. Additionally, our experiments with *N. benthamiana* suggest that FgEps1 may have a function in inhibiting host immunity. However, further research is necessary to fully understand how FgEps1 acts as a secretory virulence factor to regulate the host immune response. This research has the potential to reveal the pathogenic mechanism of *F. graminearum*, leading to the development of new strategies for the prevention and treatment of FHB.

## Figures and Tables

**Figure 1 jof-09-01009-f001:**
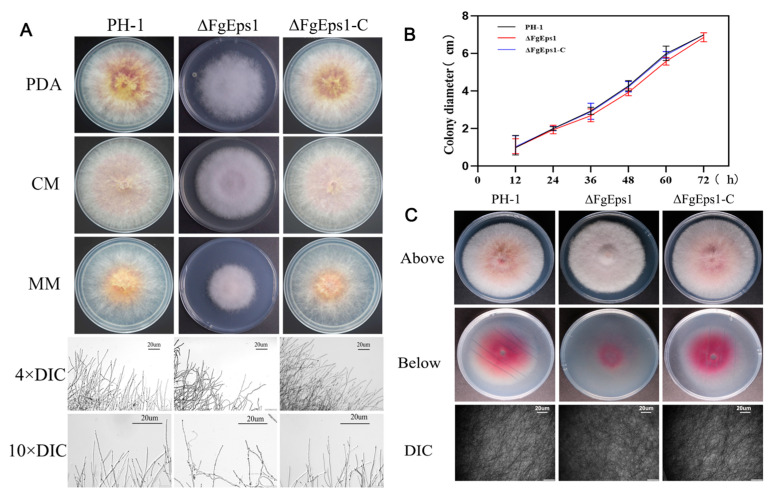
Effect of *FgEps1* deletion on mycelial growth and penetration of *F. graminearum*. (**A**) WT PH-1, ∆*FgEps1* deletion mutant, and ∆*FgEps1*-C complemented strains were grown on PDA, complete medium (CM), and minimal medium (MM), respectively, at 25 °C for 3 days. Hyphal growth at the edges of PH-1, ∆*FgEps1*, and ∆*FgEps1*-C colonies is shown (bar = 20 μm). (**B**) Change in growth rate of wild-type PH-1, deletion mutant ∆*FgEps1*, and complemented ∆*FgEps1*-C per 12 h on PDA medium. (**C**) Penetration ability of 3-day-old PH-1, ∆*FgEps1*, and ∆*FgEps1*-C strains on cellophane. Each experiment was repeated three times. The data are presented as mean ± standard deviation (SD). The asterisks indicate a significant difference.

**Figure 2 jof-09-01009-f002:**
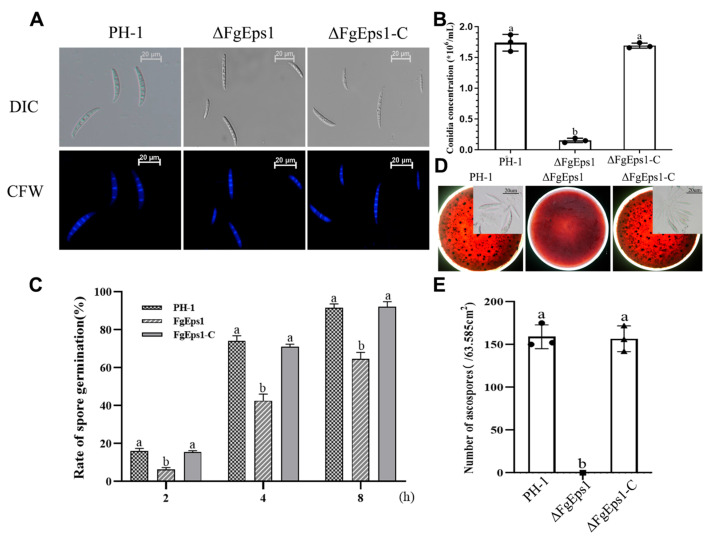
Effects of *FgEps1* deletion on asexual and sexual reproduction in *F. graminearum*. (**A**) Conidial morphology produced by 3 days incubation in CMC liquid medium is shown. (**B**) Conidia productions by *F. graminearum* strain PH-1, ∆*FgEps1*, and ∆*FgEps1*-C were measured by counting the number of conidia produced in 3-day-old CMC cultures. Bars denote standard deviations from three repeated experiments. (**C**) Conidia of all strains were incubated in YEPD liquid medium at 25 °C for 2, 4, and 8 h, and then 100 spores were examined for spore germination. (**D**) Sexual reproduction status of each strain on carrot agar medium and morphology of ascospores after dissection of sexual ascospore shells (bar = 100 μm). (**E**) The unit area of ascospores produced by each strain on each 9 cm × 9 cm carrot agar medium is shown. Each experiment was repeated three times. The same letter on the bars for each treatment represents no significant difference at *p* < 0.05.

**Figure 3 jof-09-01009-f003:**
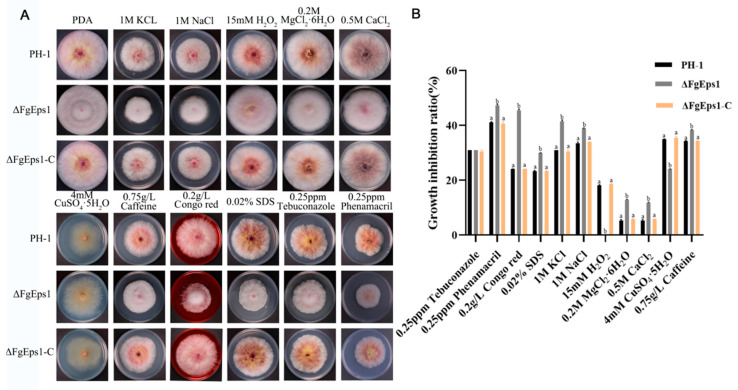
Sensitivity of Δ*FgEps1* to different stress factors. (**A**) Growth phenotypes of WT PH-1, ∆*FgEps1* mutant, and ∆*FgEps1*-C complement in PDA medium with 0.25 ppm cyhalothrin, 0.2 g/L Congo red, 0.02% SDS, 1 M KCl, 1 M NaCl, 0.2 M MgCl_2_·6H_2_O, 0.5 M CaCl_2,_ 0.75 g/L caffeine, 15 mM H_2_O_2_, CuSO_4_·5H_2_O, and 0.25 ppm tebuconazole, respectively, and cultured at 25 °C for 3 days. (**B**) Statistical analysis of the growth inhibition rate of the strains to the above-mentioned stressors is provided. Each experiment was repeated three times. The same letter on the bars for each treatment represents no significant difference at *p* < 0.05.

**Figure 4 jof-09-01009-f004:**
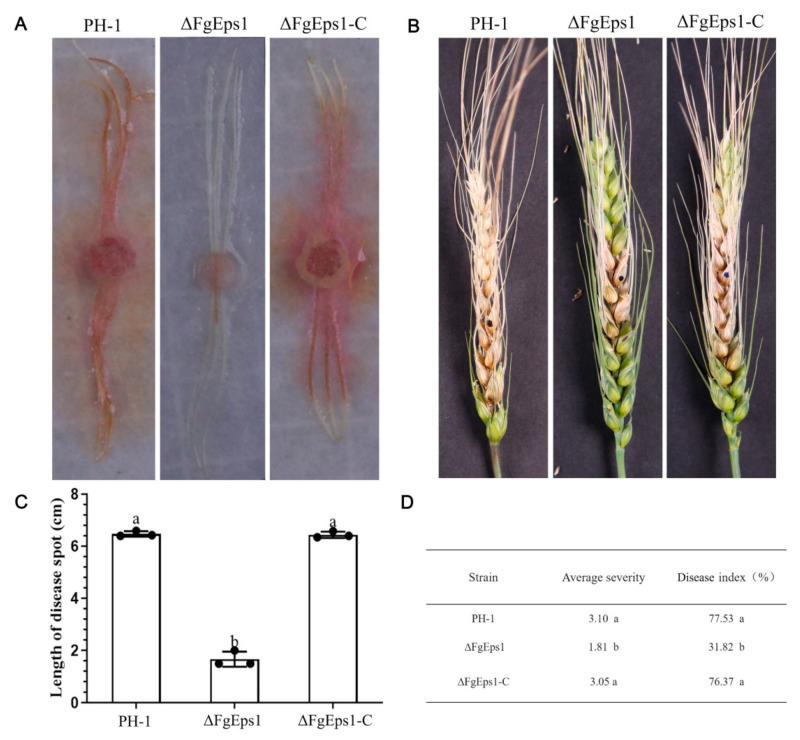
Effect of *FgEps1* deficiency on the pathogenicity of *F. graminearum*. (**A**) The infection status of maize silk inoculated with strain PH-1, ∆*FgEps1*, and ∆*FgEps1*-C is displayed. Each experiment was repeated three times. (**B**) Disease 15 days after inoculation of flowering wheat heads with conidial suspensions of PH-1, ∆*FgEps1*, and ∆*FgEps1*-C is shown. (**C**) Infection length of maize silk inoculated with strain PH-1, ∆*FgEps1*, and ∆*FgEps1*-C is presented. (**D**) Disease index and mean severity of wheat heads inoculated with strain PH-1, ∆*FgEps1*, and ∆*FgEps1*-C are shown. Means and standard deviations were calculated from three independent experiments. Values followed by the same letter are not significantly different (*p* < 0.05).

**Figure 5 jof-09-01009-f005:**
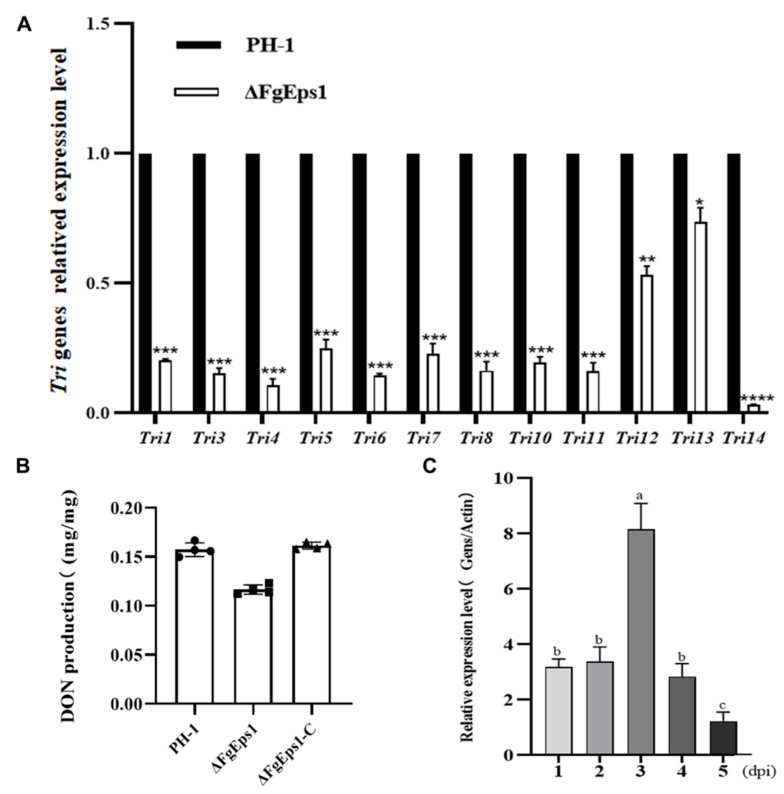
Expression levels of *FgEps1* during the period of *F. graminearum* infection and the effect on *Tri* gene clusters and DON toxin synthesis after deletion. (**A**) The relative transcription levels of 12 *Tri* genes in WT PH-1 and ∆*FgEps1* are shown. For each gene, the expression level in PH-1 was normalized to 1. Bars denote standard deviations from three repeated experiments. Significant differences compared with PH-1: * *p* < 0.05, ** *p* < 0.01, *** *p* < 0.001, **** *p* < 0.0001. Error bars represent SDs of three biological replicates. (**B**) The levels of DON produced by each strain in infected spikelets collected from inoculated wheat heads are displayed. Bars denote standard deviations from four repeated experiments. (**C**) The expression levels of FgEps*1* during *F. graminearum* infection are shown. Bars denote standard deviations from three repeated experiments. Values on the same letters following the bars are not significantly different (*p* < 0.05).

**Figure 6 jof-09-01009-f006:**
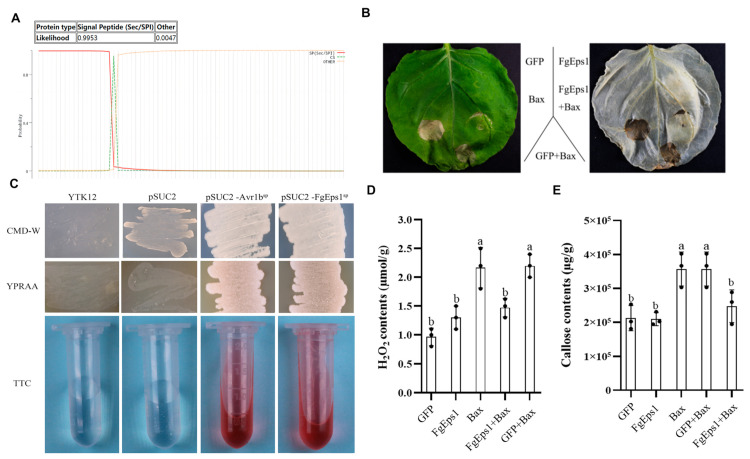
Effect of FgEps1 on H_2_O_2_ and callose accumulation in *N. benthamiana* and analysis of its signal peptide secretion function. (**A**) Signal peptide prediction of FgEps1 using the Signal Peptide Secretion Prediction Tool is presented. (**B**) FgEps1 can inhibit Bax-induced cell death in *N. benthamiana*. (**C**) The yeast YTK12 strain carrying FgEps1 signal peptide fragment fused in pSUC2 vector was able to grow in CMD-W and YPRAA medium and induce TTC red response. (**D**) H_2_O_2_ content was measured in *N. benthamiana* leaves collected for 2 days. (**E**) *N. benthamiana* leaves collected for 2 days were used to measure callose content. Bars denote standard deviations from three repeated experiments. The same letter on the bars for each treatment represents no significant difference at *p* < 0.05.

## Data Availability

The data presented in this study are available on request from the corresponding author.

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
