# Peer review of "Protein Disulfide Isomerase FgEps1 Is a Secreted Virulence Factor in Fusarium graminearum"

_jof, 2023, doi:10.3390/jof9101009_

Round 1

Reviewer 1 Report

The authors investigated FgEps1, a disulfide bond isomerase, in the fungal pathogen Fusarium graminearum, and found that FgEps1 mutation affected nutritional growth, asexual and sexual reproduction, and stress tolerance. The deletion mutant of FgEps1 resulted in reduced pathogenicity and impaired DON toxin biosynthesis. The expression of FgEps1 was detected during infection. The study showed that FgEps1 suppressed the immune response of plants.

There is some novel information in the study. The experiments primarily are well designed, results are clearly presented, and reasonable conclusions are drawn.

I have some major concerns:

Fig.1B, not significant growth difference. Therefore, it is not accurate to state: In summary, these results suggest that FgEps1 is involved in the mycelial growth and penetration ability of F. graminearum.

Should be mycelial morphology or phnotype

Fig. 2A: Please use higher magnification to show morphology.

Fig 3A, I barely can see the letters. Please increase the letter size.

How do you explain “the ΔFgEps1 strain showed increased tolerance to 15mM H2O2” but caused less disease? If the ΔFgEps1 strain showed increased tolerance to 15mM H2O2, the mutant should grow better during infection? May result in more disease?

Fig. 5A. Please list information for the time of sample collection. I assume the gene expression data from plant infection.

In Methods 2.8, Tri gene cluster expression level was tested in induction medium (TBI) liquid medium. I did not see the results. How the toxin levels in TBI? 15-DON content produced by WT, mutant, complemented mutant, should be measured and compared in toxin induction medium.

Fig. 5b: The toxin level appears to be very low in infected heads. Please doble check your toxin measurement.

In inoculated wheat heads, less toxin may be result from less disease.  DON content should be divided by fungal biomass and compared. 

Fig. 6A: one could not see much. Please modify.

Fig. 6E The co-injection also inhibited the accumulation of H2O2 and callose caused by Bax (Figure. 6D, E).

This is not accurate. The co-injection of FgEps1 did not change Bax induced callose deposition.

Some sentences need to be clarified. 

The biosynthesis of DON toxin occurs through the activity of the Tri gene cluster, in which all genes are involved except for the acetyltransferase genes, Tri101 and Tri16 [6].

This is not right. Both Tri6 and Tri101 is involved in DON biosynthesis. Please double check the reference.

inhibition of PDI gene family members can inhibit the replication of several viruses, such as HIV [31].

How: please add some explaination.

The wheat susceptible cultivar "Xiaoyan 22" was used as the susceptible cultivar.

Should be: The wheat susceptible cultivar "Xiaoyan 22" was used.

The protein coded by the FgEps1 (FGSG_06174) gene in F. graminearum comprises amino acids. Rewrite the sentence.

What do you mean “The activated wild-type strain PH-1,….”? with some kind of treatment to activate?

OD=1, you mean OD600?

Sampling was done continuously at the same time and place for 5 days after liquid nitrogen grinding..

Please clarify.

3.4 Disease index = Prevalence × Average severity ×100%; Average severity = Σ (severity× number of diseased spikes) / total number of diseased spikes investigated.

This should be in methods.

2.7 Maize silk assays should include more details or a reference.  How did you collect maize silk, what were the stage of cob development, how old was the fungal culture for inoculation?  How many samples were included?

For wheat pathogenies assays:  25 wheat spikes were inoculated with each strain at 20 µL each time. After 15 days, three wheat spikes of each strain were tested for DON toxin.

Why only tested 3 spikes for DON, not all inoculated spikes?  It should test all of them.

Please go through it carefully to make corrections.  

Abstract: FgEps1 suppressed the immune…and promoted fungal infection.

N. benthamiana, some not italic,

 1 × 105 conidium /mL should be 105

Some sentences need to be improved.

Reviewer 2 Report

Journal of Fungi Manuscript Review

Title: Protein Disulfide Isomerase FgEps1 is a Secreted Virulence Factor in Fusarium graminearum

Authors: Kouhan Liu, Xintong Wang, Ying Li, Yifeng Shi, Yanyan Ren, Aolin Wang, Bingjie Zhao , Peng Cheng* and Bao-tong Wang*

This paper shows that FpEps1 encoding a protein disulfide isomerase is essential for the pathogenicity of F. graminearum. In the manuscript, the authors generated the FgEPS1 mutant to study the function of this gene. The results support the conclusion that FpEps1 plays a crucial role in the growth and pathogenicity of F. graminearum.

The introduction is well-written, and sufficient background information is provided. The methods used are described in detail and easy to follow.  There are improvement could be made in the result and discussion. The explanation for Figure 1B and Figure 5 are lacking in the result section. Several paragraphs in the discussion read like results and could be improved.

Specific comments were provided in the PDF file attached.

The quality of English is high.
